# Real-World Study: A Powerful Tool for Malignant Tumor Research in General Surgery

**DOI:** 10.3390/cancers14215408

**Published:** 2022-11-02

**Authors:** Liang Zhang, He Li, TianFu Wang, RuiXin Wang, Long Cheng, Gang Wang

**Affiliations:** 1Department of Pancreatic and Biliary Surgery, The First Affiliated Hospital of Harbin Medical University, Harbin 150000, China; 2Key Laboratory of Hepatosplenic Surgery, Ministry of Education, The First Affiliated Hospital of Harbin Medical University, Harbin 150000, China; 3Department of Centric Operating Room, The First Affiliated Hospital of Harbin Medical University, Harbin 150000, China

**Keywords:** real-world study, general surgery, malignant tumor, treatment

## Abstract

**Simple Summary:**

In the context of today’s “accurate” medical care, the disadvantages of traditional randomized controlled trials have become increasingly prominent. At the same time, real-world study is more and more respected by people. Therefore, the purpose of this paper is to review the application of real-world study in the treatment of malignant tumors in general surgery in the past three years, to provide real-world evidence for clinical diagnosis and treatment. More importantly, while summarizing the existing research results, this paper also puts forward the current shortcomings and prospects of real-world study, which will contribute to further research, to provide help for the treatment of malignant tumors in general surgery.

**Abstract:**

Real-world study (RWS) is a method to draw conclusions by collecting and analyzing real-world data under a real clinical background. Compared with traditional randomized controlled trials (RCTs), RWSs are favored by clinicians because of their low cost and good extrapolation. In recent years, RWS has made remarkable achievements in the field of general surgery, especially in the drug treatment of advanced malignant tumors. Therefore, to further understand the main contents of the existing RWS and the application prospect of RWS in the future, this paper systematically reviews the clinical application of RWS in malignant tumors in general surgery in the past three years.

## 1. Introduction

With the development of modern medicine, people’s understanding of all kinds of diseases is deepening, and the means of treating diseases are changing with each passing day. How to choose a “cost-effective” treatment scheme suitable for patients has become the focus of clinical diagnosis and treatment. In this context, evidence-based medicine has undergone unprecedented development. The 21st century is not only a digital and information age but also an era of “data-guided decision-making”. Medical institutions everywhere have accumulated a large amount of clinical data and have established a variety of large-scale database platforms. This makes it possible to conduct clinical research based on real-world data (RWD). At the same time, the evidence needed for medical decision-making is also showing a trend in diversification and rapid growth. Therefore, real-world study (RWS) has undoubtedly become one of the research methods of most concern in recent years.

The concept of RWS was first put forward on the basis of pragmatic clinical trials to evaluate the actual effect of interventions [1]. In 1993, Kaplan et al. formally proposed RWS in a paper on the therapeutic effect of ramipril on hypertension [2]. RWS is not a new type of research but includes almost all observational and experimental research designs [3]. RWS belongs to the category of effect study [4], which specifically refers to the non-random selection of treatment measures and long-term evaluation on the basis of a wide range of subjects and large samples, and pays attention to the outcome indicators of clinical significance to further evaluate the external effectiveness and safety under real-world conditions [5]. Other concepts closely related to RWS are RWD and real-world evidence (RWE). The former is mainly defined as data on patients’ health status and/or health care collected in a nonrandomized controlled trial environment [6,7]. The latter mainly refers to evidence collected in clinical care and family or community environments rather than in research-intensive or academic environments [6]. Thus, RWS can be understood as the research that produces RWE by collecting and analyzing RWD. The novelty of this paper lies in it being the first systematic review of RWS related to malignant tumors in general surgery in the past three years, through the analysis of research conclusions to guide clinical treatment. This paper is different from other reviews in that most reviews on malignant tumors in general surgery only focus on the conclusions drawn by traditional RCT. While summarizing the conclusions of RWS in the past three years, this paper emphasizes the application status of RWS itself as a new research method in the study of malignant tumors in general surgery. In addition, this paper also puts forward views on the shortcomings of the existing RWS and the next development direction.

## 2. RCT and RWS

Randomized controlled trials (RCTs) have always been recognized as the gold standard for evaluating interventions and have been at the top of the pyramid of evidence-based medicine for a long time. However, traditional RCTs have shortcomings. First, due to strict conditions, the object of study is often in an “ideal state”, which is inconsistent with reality. Second, patients lack full understanding of it, and their compliance is poor, which leads to the unavailability of the research data. Third, the cost of the research is high, but the extrapolation of the experimental results is poor. Therefore, compared with RCTs, RWSs have obvious advantages. First, in the selection of research subjects, compared with the subjects strictly screened by RCT, the objects of RWS are often unrestricted or relaxed, and the sample size of RWS is much larger than that of RCT. This not only effectively avoids RCT selection bias but also greatly improves the external authenticity of the study, making the study more in line with clinical practice. Second, in terms of intervention measures, compared with the strict standardized intervention measures of RCTs, RWS tends to give no intervention or fewer intervention measures. This is not only close to clinical practice but also improves the compliance of patients, which is convenient for long-term follow-up in the later stage of the study. Finally, in terms of research cost, the single sample data cost of RWS is much lower than that of RCT, and both the research object and the research problems come from clinical practice, such as among the existing treatment schemes received by patients, which has the best treatment effect and the best prognosis. In addition, the novelty of RWS is that, compared with the various “ideal states” of traditional RCT, RWS puts the whole process of research in a “real clinical environment” for the first time. Its research background, objects and data all come from clinical rather than traditional “experimental data”. In short, the important value of RWS is that it can make up for the shortcomings of traditional clinical trials, using the existing diversification of big data to produce a variety of evidence to meet the current needs of medical staff for a variety of evidence. Additionally, RWE can be used in a wide range of applications, both observational and planned intervention studies [6].

However, there are no shortcuts to scientific research, and each research method has its own advantages and disadvantages. While RWS has obvious advantages, it also has some limitations and shortcomings. First, the problem of data collection. RWD has a wide range of sources, various forms and large orders of magnitude, so it needs strong technical support when collecting data; for example, it will be time-consuming and laborious to complete it only by manpower. In addition, the collection of electronic data may be inconsistent with reality, and improper handling of missing data elements will also lead to a decrease in statistical validity and the ability to answer research questions [8]. The second aspect is the data quality. While RWS solves the extrapolation problem of RCT, bias and confusion are the problems that every RWS must face and address. Because there are many interference factors in an RWD, its completeness, reliability and accuracy are often not high [3]. The third is privacy and data security. When using retrospective data, the most important thing is the security of data. Whether researchers have access to some private data is still controversial, and there is still a lack of relevant data access policies [9]. In short, RCTs and RWSs have their own advantages and disadvantages, as shown in Table 1. There is no “good” or “bad” distinction between the two research methods, and the two are not antagonistic but complement each other and coexist with each other.

Based on the above background, this study retrieved 1869 full-text articles in PubMed with the search of “(“real world study” [Title/Abstract]) OR (“real world research” [Title/Abstract])” and calculated the number of articles published in each year. As shown in Figure 1, the number of RWS-related papers published has been increasing year by year since 2015, and the number of published papers has increased sharply to 670 in 2021 alone. Therefore, this review summarizes the application of RWS in malignant tumors in general surgery in the past three years.

## 3. Breast Cancer (BC)

As the most common female cancer in the world, BC accounts for 30% of all newly diagnosed cancers [10]. Approximately 20–30% of the early patients developed metastatic breast cancer (MBC) during follow-up, while 5–10% of female patients have primary MBC [11]. At present, for advanced hormone receptor-positive (HR+) human epidermal growth factor receptor type 2-negative (HER2-) BC patients, endocrine therapy (ET) is often recommended by the guidelines when they have no resistance to visceral disease or ET [12]. Fulvestrant has proven its clinical effectiveness in previous trials as a primary or second-line treatment of HR+/HER2- postmenopausal advanced BC [13]. Recently, a retrospective, observational study involving 303 HR+/HER2- patients with advanced BC found that fulvestrant is an effective, safe and well-tolerated ET for HR+/HER2- MBC [13], and it can also be used as a maintenance therapy after first-line palliative chemotherapy. A cohort study found that for HR+/HER2- BC patients, compared with using fulvestrant alone, the combination of palbociclib and fulvestrant increased the risk of clinical adverse events such as neutropenia, leukopenia and thrombocytopenia [14]. At the same time, the risk of acute liver injury was also increased. Before that, an RWS in five European countries found that for advanced patients, even after receiving first-line ET-based regimens, their health-related quality of life was still poor, and fatigue, pain, insomnia and activity disorders were their main problems [15]. However, a recent RWD has demonstrated that ET plus cyclin-dependent kinase 4/6 inhibitors (CDK4/6i) still represents the best first-line treatment for HR+/HER2- MBC compared to ET or chemotherapy alone [16]. In the prediction of the efficacy of fulvestrant, the study found that the best duration of previous ET was the main predictor. In addition, no liver metastasis, simple bone metastasis, first-line administration and sensitivity to previous ET were also good predictors of efficacy [17]. In the chemotherapy cycle, retrospective studies found that for lymph node-negative and HER2- BC patients, four cycles of docetaxel and cyclophosphamide adjuvant chemotherapy are the most beneficial in the long-term survival of patients [18].

It has been demonstrated that trastuzumab can significantly improve clinical efficacy and lay a foundation for modern biological targeted therapy of HER2+ breast cancer [19]. However, most patients experience disease progression during or after trastuzumab treatment [20,21], so additional intervention is usually needed. For patients with HER2+ MBC, a retrospective study found that tyrosine kinase inhibitors (TKIs) combined with chemotherapy were effective in patients who had previously received trastuzumab [22]. In the same year, similar conclusions were found in two studies involving patients with HER2+ MBC; lapatinib-based therapy was effective in patients (even in patients pretreated with trastuzumab) and the combination of lapatinib and capecitabine should be recommended because of its good efficacy, convenience and tolerance [23,24]. T-DM1 is an antibody drug conjugate that combines trastuzumab with emtansine via a stable thioether linker [25]. In a single-center retrospective study, T-DM1 was also found to be effective, safe and tolerable in patients with HER2+ MBC, confirming its effectiveness in routine clinical practice [25]. In terms of drug side effects, an RWS evaluating the cardiac safety of intravenous trastuzumab in patients with HER2+ BC found that the incidence of symptomatic congestive heart failure and cardiac death was consistent with the conclusions of randomized clinical trials [26]. The associated cardiotoxicity may be reversible and mild, and standard cardiac drugs may address these cardiac events. In addition, a retrospective study also found that patients with HR+/HER2- were treated with drugs that led to prolonged cardiac repolarization (prolonged QT intervals), underscoring the importance of individualization and risk assessment [27].

Pyrotinib is a novel pan-HER2 TKI, and its efficacy in HER2+ MBC has become the focus of recent research. In a retrospective study involving 94 patients, pyrotinib showed good results in the treatment of HER2+ MBC patients, especially in patients who did not take lapatinib, and showed some activity in patients treated with lapatinib [28]. In a prospective study of 141 BC patients with HER2+, it was also confirmed that pyrotinib was effective and well tolerated for HER2+ BC [29]. For HER2+ MBC patients who failed trastuzumab and lapatinib treatment, studies have shown that pyrotinib is equally effective and has fewer adverse reactions in patients, especially those who benefit from previous treatment with lapatinib or those who do not have distant metastasis [30]. In addition, in a cohort of 168 patients, retrospective studies found that surgery or radiotherapy combined with pyrotinib therapy significantly improved overall survival (OS) in patients with HER2+ MBC [31], and it may have advantages in preventing brain metastasis, as well as good efficacy in patients with brain metastasis. For MBC patients with brain metastasis, systemic drug therapy is beneficial because it effectively delays local intracranial therapy [32]. At the genetic level, an RWS showed that FGFR, TP53 and FLT1 genetic aberrations, as well as positive HER2, can increase the risk of brain metastasis in MBC patients, and FGFR genetic aberration alone predicted poor prognosis [33]. An RWS study on pyrotinib in 2021 reached the same conclusion as the above study, and it is worth noting that the combination of pyrotinib and trastuzumab showed an advantage in progression-free survival, even in patients resistant to trastuzumab [34]. The study also points out that pyrotinib-based treatment may be the first choice for patients with brain metastasis, especially in combination with brain radiotherapy [34].

Triple-negative breast cancer (TNBC) with negative estrogen receptor, progesterone receptor and HER2 has been paid more and more attention because of its strong invasiveness and high risk of recurrence and death [35]. A multicenter RWS discovery showed that platinum-based chemotherapy (PBC) is superior to non-PBC doublets in terms of efficacy and manageable toxicity in clinical treatment for advanced TNBC patients [36].

In the neoadjuvant chemotherapy and surgical treatment of BC, an RWS study found that epirubicin/cyclophosphamide with weekly paclitaxel–trastuzumab has more advantages in both pathologic complete response (PCR) and disease-free survival (DFS) [37]. Compared with traditional anthracycline, pegylated liposomal doxorubicin (PLD) has similar efficacy, but its cardiotoxicity is significantly reduced [38]. The study found that the efficacy of PLD is similar to that of epirubicin in neoadjuvant and adjuvant chemotherapy for BC, but the toxicity is relatively lower [39]. This shows that PLD may have greater research value in the future. A retrospective study of the real world in China found that axillary surgery is unnecessary in patients with negative axillary lymph nodes who achieve breast PCR (T0/Tis) or initial positive axillary lymph nodes who achieve T0 (not Tis) [40].

In terms of prognosis and tumor recurrence, it is well known that the prognosis of stage III BM (any TN3M0) is poor. Recently, a single-agency RWS found that there was still a difference in prognosis among N3a, N3b and N3c. N3c patients had the worst clinical prognosis, while N3b patients had a better prognosis than N3a patients [41]. A retrospective RWS found that changes in Ki67 can help predict the prognosis of patients with luminal type B BC, providing a measurement of neoadjuvant chemotherapy efficacy and assisting in further clinical decisions [42]. An RWS observed that HER2+ BC patients with PCR of neoadjuvant pertuzumab plus trastuzumab still have a certain probability of recurrence [43]. In this regard, after receiving neoadjuvant therapy to achieve PCR, we can consider the combination of trastuzumab and pertuzumab to replace the regimen of trastuzumab alone to improve the prognosis of patients [43]. In addition, a longitudinal RWS of BC patients showed hospital-based comprehensive medical plans can relieve the fatigue of BC patients during treatment [44]. In a recent study, the evaluation of RWD showed that the application of the concept of multimodal therapy can significantly enhance the internal coherence and resilience of BC patients [45].

Finally, in terms of screening strategies for BC in high-risk women, a study based on real-world populations found that ultrasound alone or mammography screening after ultrasound screening was a sensitive, cost-effective screening method, and its effect was better than that of mammography alone [46]. For patients with MBC, an effective monitoring strategy is very important to monitor the development of the disease and evaluate the effectiveness of treatment. A recent retrospective study of 382 patients with MBC analyzed different monitoring strategies and found that serum tumor markers (STM) and CT/MRI were the most commonly used indicators for monitoring [47], but the optimal frequency of monitoring was not clearly defined.

## 4. Hepatocellular Carcinoma (HCC)

Sorafenib is an oral multikinase inhibitor. RCTs have confirmed that it can significantly prolong the overall survival time of patients with advanced hepatocellular carcinoma (aHCC) to 2.8 months [48,49]. A single-center real-world cohort study not only confirmed that sorafenib can improve OS in a real clinical environment but also indicated that careful selection of patients treated with sorafenib is essential to optimize treatment results [50]. In addition, a multicenter RWS pointed out that sorafenib is equally effective and safe for patients with HCC under dialysis, and its efficacy is comparable to that of the nondialysis population [51].

As an alternative treatment for patients with aHCC, lenvatinib has been approved worldwide [52]. In an RWS, lenvatinib has been shown to be more effective than sorafenib in patients with unresectable HCC (uHCC), and there is no significant difference in safety between them [53]. A retrospective study not only confirmed that lenvatinib has a good curative effect on aHCC patients but also showed that baseline characteristics, changes in serum biomarkers and gene sequencing may be the key to predicting lenvatinib responses [54]. A multicenter analysis in Korea also confirmed that lenvatinib has a good curative effect on aHCC patients, and the treatment is well tolerated by patients [55]. Further studies have found that for patients with uHCC, lenvatinib-based combination therapy (lenvatinib plus PD-1 antibody plus transcatheter arterial chemoembolization (TACE) or hepatic arterial infusion with drug filtration (HAIF), lenvatinib plus PD-1 antibody, lenvatinib plus TACE or HAIF) is more effective than lenvatinib monotherapy [56]. When uHCC patients become resistant to lenvatinib, sorafenib as a second-line treatment may be effective and may not worsen the liver reserve [57].

In a phase III clinical trial, regorafenib demonstrated survival benefits for uHCC patients [58]. A recent RWS confirmed that regorafenib was effective and well tolerated in uHCC patients who had previously failed treatment with sorafenib, especially in patients with lower baseline alpha-fetoprotein (AFP) levels (AFP < 400 ng/mL) or with an early AFP response [59]. HCV infection, albumin–bilirubin (ALBI) grade, macrovascular invasion, hand–foot skin reaction (HFSR) and baseline AFP were independent prognostic factors in patients treated with regorafenib [59]. A previous multicenter retrospective study found that regorafenib combined with TACE as a second-line treatment regimen was beneficial and well tolerated in uHCC patients who failed first-line treatment [60]. The prognosis of patients with controlled disease after TACE combined with regorafenib treatment may be better. In addition, tumor diameter, AFP level and dose of regorafenib and best response to regorafenib are all related to OS and progression-free survival (PFS) of patients [60].

Hepatic resection is suitable for early-stage (stage 0 and A) HCC patients, while TACE is more recommended for intermediate-stage HCC patients, according to the Barcelona Clinic Liver Cancer (BCLC) staging system [61]. A multicenter, retrospective study found that in the real clinical environment, raltitrexed-based TACE can prolong the OS of patients with aHCC, and it is safe and tolerable [62]. A cohort study found that for intermediate-stage HCC, when LDH levels were >192 U/L, the therapeutic effect of hepatic resection was better than TACE; TACE may be more suitable for patients with LDH levels ≤ 192 U/L [63]. It is worth noting that when HCC patients receive radical hepatectomy, not all patients can benefit from postoperative adjuvant TACE; patients ≥ 50 years old, tumor size > 5 cm or CN stage Ib/IIa are strongly recommended to receive postoperative adjuvant TACE therapy [64]. The treatment of aHCC is often multimodal, and it is difficult to achieve disease control with only a single therapy. A retrospective analysis of aHCC found that hepatic artery infusion chemotherapy combined with anti-PD-1 immunotherapy and TKIs was effective and safe for late-stage patients [65]. In addition, retrospective studies have found that both primary HCC and metastatic HCC and high-intensity focused ultrasound (HIFU) are effective and safe treatments, and HIFU may also be the best treatment for uHCC [66].

In terms of prognosis, for HCC patients, a visual analog scale ≥ 5, longest tumor diameter ≥ 5 cm, extrahepatic metastasis and portal vein invasion all represent worse prognosis [66]. Among aHCC patients treated with ramucirumab, an RWS found that patients with a modified albumin–bilirubin (mALBI) score of 1 to 2a had a better prognosis [67]; in addition, an early change in the AFP level may also be helpful in predicting the prognosis of patients. For N0M0 patients undergoing radical hepatectomy, an RWS found that older, Black, male patients, higher tumor stage, larger tumor, traditional radiotherapy and higher AFP levels were often associated with poor prognosis [68]. In short, in the current real clinical environment, it is generally believed that tumor size, clinical stage, tumor invasion extent, treatment regimen and AFP level are closely related to the prognosis of patients.

## 5. Colorectal Cancer (CRC)

Fluorouracil-based chemotherapy combined with targeted therapy is the first choice for patients with advanced colon cancer; surgical treatment is considered only when complications such as bleeding, perforation and obstruction occur [69]. An RWS found that bevacizumab combined with first-line chemotherapy is equally safe and effective in elderly patients with metastatic colorectal cancer (mCRC), and its benefit/risk balance is similar to that in younger patients [70]. A recent retrospective study also pointed out that standard optimal chemotherapy regimens (e.g., 5-FU-based adjuvant chemotherapy) can also significantly improve OS and are well tolerated in elderly patients with CRC [71]. All of these findings suggest that we should be more proactive in the clinical treatment of elderly patients, and simple old age is not a factor that hinders the treatment of elderly patients. A retrospective cohort study found that in the real clinical environment, S-1 is similar to capecitabine’s OS rate, but the incidence of adverse events is lower; therefore, S-1 may be the first choice for Chinese patients with mCRC [72].

An RWD-based study found that when unresectable mCRC patients received standard first-line cetuximab-based treatment, maintenance therapy containing cetuximab could significantly improve survival and improve their quality of life with less toxicity; therefore, cetuximab can be used as an effective and safe maintenance treatment drug for mCRC patients [73]. Regarding the treatment cycle of cetuximab, an RWS found that compared with the 250 mg/m^2^ once weekly (Q1 W) regimen, the 500 mg/m^2^ every 2 weeks (Q2 W) regimen was not inferior to the patient’s OS and safety; therefore, cetuximab Q2 W can be used as a replacement for cetuximab Q1 W regimen because the former helps synchronize the administration of cetuximab and concomitant chemotherapy [74]. In addition, a prospective, cross-sectional, noninterventional study found that trifluridine/tipiracil (FTD/TPI) is a well-tolerated option for the treatment of refractory mCRC patients, which improves the overall quality of life of patients in real clinical settings [75].

In terms of surgical treatment, an RWS based on the SEER database shows that surgery on the primary tumor (SPT) is beneficial to improve the survival rate of patients with stage IV colon cancer, regardless of metastasis site and the number of years of diagnosis [76]. This study showed that younger age, left colon cancer site, well-differentiated and moderately differentiated tumors, T1/T2 stage, more lymph nodes examined, fewer positive lymph nodes and chemotherapy were significantly associated with better prognosis in patients receiving SPT [76]. For postoperative patients, a retrospective observational study found that for patients with high-risk stage II colon cancer, active postoperative adjuvant chemotherapy can greatly improve the 5-year OS; however, the correlation between different treatment regimens such as surveillance, single- and multiagent chemotherapy and patient survival outcomes still needs to be further evaluated in the next study [77]. With regard to the toxicity of adjuvant chemotherapy, an RWS noted that approximately 1/5 of patients were hospitalized during adjuvant chemotherapy. Patients who are older, female, have more complications and receive adjuvant CAPOX therapy are more likely to be hospitalized [78].

In terms of prognosis, a multicenter RWS found that antiangiogenic-based regimens prolonged median PFS in patients with mCRC, while the systemic inflammation score, neutrophil–lymphocyte ratio (NLR) and serum albumin could predict the efficacy of treatment [79]. In addition, the study also confirmed that patients with BRAF mutations responded poorly to treatment than wild-type (WT) BRAF tumors, resulting in a worse prognosis [79]. A retrospective study also confirmed that in mCRC patients, WT patients have higher OS [80]. Second, the OS of patients with left tumors, primary tumor resection and metastasis was significantly better than that of patients with right tumors, unresectable primary tumors and metastasis [80]. In addition, the original nomograms constructed according to the clinicopathological characteristics of patients can help clinicians predict the specific distant metastasis site and OS of patients, predict the prognosis of patients and help to individualize the evaluation of patients after operation [81]. As nomograms showed, positive CEA levels are usually associated with lower OS rates and higher distant metastasis rates in patients with CRC; the larger the tumor, the higher the T and N stages, the higher the risk of lung, liver, bone and brain metastasis, leading to a worse prognosis, while younger patients and tumors located in the left colon are better indicators of prognosis. In addition, this nomogram also verified that signet ring cell carcinoma (SRCC) with lower tissue differentiation usually represents a worse prognosis [81].

## 6. Gastric Cancer (GC)

Apatinib, an oral receptor TKI that specifically targets vascular endothelial growth factor receptor 2 (VEGFR2), has been shown to inhibit the angiogenesis of tumors by prohibiting VEGF-promoted tumor development [82]. A prospective, multicenter observational study in the real world found that in the treatment of advanced GC, low-dose apatinib (500 mg or 250 mg per day) therapy was an effective regimen, well tolerated by patients, while regimens such as combinations with chemotherapy, including apatinib plus taxol/docetaxel, could improve the survival rate of patients [82]. An RWS has also confirmed that lower doses of apatinib therapy can bring survival benefits to patients with advanced GC, and the patients tolerated it well. Therefore, a lower dose of apatinib therapy can optimize the therapeutic effect of apatinib and minimize its side effects in clinical practice [83]. A multicenter, prospective study from China simultaneously confirmed the effectiveness and safety of apatinib in first-line treatment; it is especially pointed out that the regimen of apatinib combined with paclitaxel will bring greater survival benefits to GC patients [84].

In terms of surgical treatment, a retrospective RWS indicated that D2 gastrectomy could be a safe and effective treatment for GC patients with radiologically suspicious para-aortic lymph node metastasis (PALM) who respond well to chemotherapy [85]. Moreover, neoadjuvant chemotherapy (NAC) is feasible and safe for locally advanced GC patients; it promotes tumor degradation, eliminates potential micrometastasis and prolongs survival without increasing perioperative risk, and NAC can further improve the prognosis of patients and prevent recurrence [86]. For patients receiving D2 gastrectomy, an RWS pointed out that both postoperative adjuvant SOX (S-1 and oxaliplatin) chemotherapy and XELOX (capecitabine and oxaliplatin) chemotherapy had good survival benefits, and the survival benefits of the two regimens were similar [87]. In addition, a previous study pointed out that in the adjuvant chemotherapy regimen for patients after D2 gastrectomy, according to the Lauren classification of GC, oxaliplatin-based adjuvant chemotherapy is more effective in intestinal-type patients, compared with diffuse-type patients [88]. It is worth noting that elderly patients over 80 years old have higher perioperative morbidity, mortality and complications, as well as a high proportion of late stage or metastasis and shorter life expectancy [89]. As a result, it seems that the survival benefits of surgical treatment are not significant. However, for elderly patients with proximal GC who are recommended for surgical resection, a 2019 RWD-based study showed that less invasive surgery (such as small-scale lymph node dissection) might be the optimal treatment for elderly patients, and extensive surgery has no survival benefits for patients [90]. Finally, the prognosis of metastatic GC is usually poor. A retrospective study showed that metastatic GC patients receiving multimodal treatment (such as palliative surgery combined with radiotherapy and chemotherapy) had greater survival benefits than those receiving systematic chemotherapy alone [91]. Regarding prognosis, the degree of tumor differentiation, the level of serum CA199, the extent of previous resection and the choice of operation are all important prognostic factors affecting the survival rate of patients [91].

## 7. Issues and Prospects

As seen from the above research, RWS at home and abroad is mainly used to evaluate the efficacy and safety of drugs in real clinical environments. The main “hot” drugs and some references related to malignant tumors are listed in Table 2. It can be seen that in the past three years, RWS has made remarkable achievements in the drug treatment of malignant tumors. However, there are still some problems and deficiencies in the existing RWS.

The first limitation is the deficiency of the RWS research methods. Because part of the RWS is a retrospective study, it cannot be prospectively randomly grouped. Therefore, the conclusions of the study will be subject to some limitations, including the lack of some clinical factors, such as combination therapy, as well as possible selection bias and omissions and errors in related data. In prospective studies, because the research data are also obtained from observational studies, potential information biases or incomplete data are inevitable. Moreover, because the research background is in the real clinical environment, patients’ choice of treatment plan is more affected by nonmedical factors such as treatment cost and length of stay. This not only increases the selection bias but also inevitably affects the accuracy of the conclusion. In addition, the lack of a single center and sample size not only brings deviation to the research conclusion but also reduces the credibility of the conclusion. Therefore, to obtain more reliable evidence, multicenter prospective clinical trials will be necessary. In addition, many of the studies mentioned above are regional; that is, the study is limited to one region or country (such as Japan, South Korea, Europe, etc.). Due to ethnic differences, the research evidence or conclusions of many Western countries may not be applicable to Asia, and the high-risk factors and surgical indications of some diseases in Asia may also be different from those in Europe. This leads to the limitations of some research conclusions, which requires us to interpret these research results carefully.

The second is the deficiency of the research content. In related research on BC, the definition of “PCR” may be different in different studies, which indirectly leads to different conclusions. For patients with HER2+ and brain metastasis, it was found that pyrotinib combined with radiotherapy/surgery could improve OS [31]; however, the mechanism of pyrotinib is not clear, and further research is needed to clarify it [92]. In terms of MBC monitoring, the best monitoring strategy remains unclear and controversial. The conclusions drawn from clinical trials are likely to not meet the clinical needs of the real world. However, the applications based on liquid biopsy provide us with a new idea; for example, to better monitor the response to anticancer treatment through analysis of ctDNA and/or CTCs [47]. In a study of HCC, it was found that most Chinese patients had concomitant hepatitis B virus (HBV)-related HCC [54]. For patients with non-HBV infection, it is not known whether the related drugs have the same effect and whether it is necessary to formulate individualized treatment measures for patients with non-HBV infection. With regard to the RWS of HIFU, it was found that HIFU monotherapy was effective in patients with HCC [66]. However, it is not known whether HIFU-based combination therapy will bring greater survival benefits to patients, so more RWS is needed to further evaluate the effectiveness of HIFU-based combination therapy in a real clinical environment. As one of the important methods for the treatment of HCC, liver transplantation has always attracted people’s attention. Immunosuppressants with antirejection effects have brought a new dawn in the field of liver transplantation. However, one of the disadvantages of immunosuppression is that it reduces immune rejection and increases the risk of HCC recurrence and other complications, such as renal insufficiency [93]. Therefore, the ideal immunosuppressant should have minimum nephrotoxicity and no tumorigenicity. A prospective, multicenter study evaluated the clinical results of early conversion to sirolimus-based regimens in patients with HCC after liver transplantation. It predicts that the sirolimus conversion regimen will provide patients with survival benefits and better quality of life at the same time [93]. In real clinical studies of advanced GC, surgical treatment has been shown to significantly improve OS in patients with early, resectable GC, even in elderly patients ≥ 80 years old [90]. However, the need for chemotherapy or radiotherapy for elderly patients after surgery is still controversial. On the one hand, RWD-based studies have confirmed that the benefits of additional radiotherapy and chemotherapy after surgery are limited; on the other hand, the potential toxicity of radiotherapy and chemotherapy and the resulting decline in quality of life in elderly patients need to be considered [90]. Therefore, more prospective experiments are needed to further assess its related risks and benefits. In addition, there is still much debate about the need for palliative surgery for patients with metastatic stage IV GC. A retrospective study found that patients who received palliative surgery survived longer than those who received chemotherapy or radiotherapy alone [91]. However, this result is likely to be caused by selection bias; that is, patients who choose surgery tend to be in a better general state or local state of the disease. Therefore, in the next study, the efficacy and safety of palliative resection in stage IV GC patients should be further evaluated in large-scale prospective RCTs. In the choice of postoperative adjuvant chemotherapy, some researchers believe that adjuvant chemotherapy can be further selected according to the Lauren classification of GC patients, while for diffuse-type GC, a docetaxel-based regimen may be the focus of future research [88].

In addition, at present, most studies at home and abroad are evaluating the effectiveness and safety of drugs, but in this process, the evaluation of drug resistance is often ignored. Once patients have drug resistance problems, whether there is a suitable alternative is worthy of further study. For example, a retrospective study of HR+/HER2- MBC found that when patients become resistant to CDK4/6i as a first-line treatment, ET combined with other targeted molecules may become a second-line treatment [16]. In the aspect of measuring the effectiveness of drugs, the existing studies basically measure the effectiveness and safety of drugs from various clinical indicators, such as PFS and OS. Therefore, the potential safety and efficacy of drugs can be evaluated at the genetic level, such as tumor mutation burden, and some macroscopic indicators can be specified to the gene molecular level.

Finally, the existing research scope of RWS is very limited; most studies focus on the drug treatment of advanced malignant tumors, and only a few studies are in the evaluation of the scope of surgical resection and the corresponding prognosis. Therefore, in future research, we can expand the scope of the RWS and focus on the early intervention of the disease. For example, for diseases that have been clearly diagnosed, whether they need surgical intervention, the specific indications of surgical intervention, and the best time for intervention. In the early screening of malignant tumors, what are the high-risk factors that need routine disease screening, and what is the best screening method and frequency? For patients who receive surgical treatment, we can further evaluate the differences in OS and prognosis brought by different surgical methods and then help doctors make the best choice in clinical decision-making. In this process, we should also pay attention to the adverse effects of selection bias, insufficient sample size and various nonmedical factors, such as the economic status of patients and local medical policies.

## 8. Conclusions and Future Perspectives

The concept of RWS is not put forward for the first time only in recent years. However, the number of RWSs has increased sharply in the past three years, which is largely due to the support of Internet “big data” and the development of evidence-based medicine. RWS further confirmed the experimental conclusion of RCT with RWD in the real clinical environment. Therefore, the significance of RWS is to make use of the advantages of “big data” to make up for the shortcomings of traditional RCT. At present, RWS has made great achievements in the drug treatment of malignant tumors in general surgery. In the future, we can establish multiple-regional large “clinical databases” worldwide to collect and analyze RWD. The goal is to validate the conclusions of RCTs in the clinic, and then help us find better solutions for the management of human health problems.

## Figures and Tables

**Figure 1 cancers-14-05408-f001:**
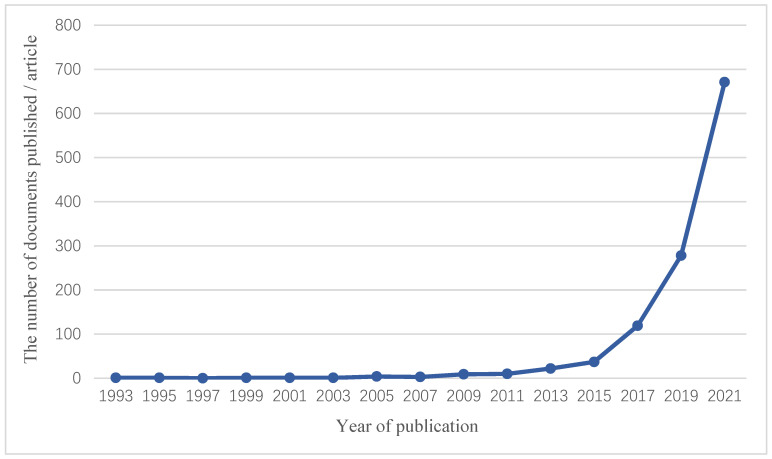
Number of RWS-related articles published over the years.

**Table 1 cancers-14-05408-t001:** Comparison between RWS and RCT.

Comparison Catalogue	RCT	RWS
Research environment	Ideal experimental conditions	Real clinical environment
Research and design	Randomized controlled trial	Observational study
Research object	Strict screening	Unrestricted condition
Number of samples	Usually small	Usually larger
Intervention measures	Strict intervention	No intervention
Compliance	Poor	Good
Interfering factors	Less	More
Cost input and extrapolation	Large cost input and poor extrapolation	Low cost input and good extrapolation

**Table 2 cancers-14-05408-t002:** RWS for drug treatment of malignant tumors.

Disease	Drug	Sample Size/Example	Research Type	Year	References
BC	Fulvestrant	303	Treatment	2020	[11]
	Palbociclib	2445	Treatment	2021	[12]
	Lapatinib	92	Treatment	2020	[21]
	Lapatinib	112	Treatment	2020	[22]
	T-DM1	15	Treatment	2019	[23]
	Pyrotinib	94	Treatment	2021	[24]
	Pyrotinib	141	Treatment	2021	[25]
	Pyrotinib	105	Treatment	2021	[26]
	Pyrotinib	168	Treatment	2021	[27]
	Pyrotinib	218	Treatment	2021	[29]
	PLD	1213	Treatment	2021	[32]
	Trastuzumab	3733	Treatment	2019	[33]
	Pertuzumab	217	Prognosis	2021	[41]
HCC	Sorafenib	115	Treatment	2020	[48]
	Sorafenib	6156	Treatment	2020	[49]
	lenvatinib	98	Treatment	2020	[51]
	lenvatinib	54	Treatment	2020	[52]
	lenvatinib	92	Treatment	2020	[53]
	Sorafenib	13	Treatment	2020	[55]
	Regorafenib	86	Treatment	2022	[57]
	Regorafenib	38	Treatment	2021	[58]
	Ramucirumab	16	Prognosis	2021	[65]
CRC	Bevacizumab	402	Treatment	2020	[68]
	S-1	1367	Treatment	2020	[70]
	Cetuximab	177	Treatment	2021	[71]
	Cetuximab	2943	Treatment	2020	[72]
	FTD/TPI	105	Treatment	2020	[73]

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
