# Peer review of "Real-World Study: A Powerful Tool for Malignant Tumor Research in General Surgery"

_cancers, 2022, doi:10.3390/cancers14215408_

Round 1

Reviewer 1 Report

This paper brings forth the usefulness of real-world studies, a concept that is not new, but has lost some ground in the scientific world to RCTs in the past decades. 
The introduction provides enough information on the topic and is well-structured, with adequate references. 

The comparison between RCT and RWS is very welcome, very clear and extremely accurate. The methodology for the data shown in figure 1 is adequately described and the simplified version presented in Table 1 is also very useful. 

The analysis performed on multiple tipes of malignancies (the most common in general surgery) is a more in-depth comparison of the two, but at the same time underlines the usefulness of using both types of study, as results of RCTs (highly valuable in terms of statistical significance and level of evidence) can be put into practice which, in turn, may be better suited to a RWS type analysis. 

The last part of the article reiterates the shortcomings of RWS, but with the increased popularity of this research method a standardization may ensue that would produce higher-quality results. 

The article is very good, it was truly pleasurable to read and the data is presented in a very clear and non-biased manner. I would like to congratulate the team and look forward to reading more articles on this topic. 

Author Response

We are very grateful to the reviewers for giving us such a high opinion of this review. In the coming time, we will keep up our efforts to follow up on RWS related to malignant tumors in general surgery and hope to bring more high-quality articles. Finally, we once again express our heartfelt thanks to the reviewers for their carefulness and responsibility.

Reviewer 2 Report

I would like to congratulate the authors for reviewing all the literature data available. 

Further studies are needed for evaluate similar data worldwide. 

Author Response

We are very grateful for the reviewer’s carefulness and responsibility for this paper. As the reviewer said, this review has systematically reviewed the application of RWS in malignant tumors in general surgery in the past three years. In the coming time, we will continue to follow up on RWS related to malignant tumors in general surgery all over the world, to further evaluate the application value of RWS.

Reviewer 3 Report

No major comments by my side.

Good work btw. 

Author Response

(The authors gave the same response as above.)

Reviewer 4 Report

Zhang et al has written a review article entitled “Real-world study: a powerful tool for malignant tumor research in general surgery”. The article is original with good concept. However there are a few places where the language is not scientific and correlation is also not very evident. These don't detract from the meaning of the text but a careful proofread could address these issues and improve the flow of the text. In my opinion, the manuscript can be published in this journal, after the authors have addressed the following comments and questions:

1.      In simple summary section, Line 15-16 is not clear

2.      Grammatical and formatting issues are there in manuscript at several places

3.      Authors are suggested to improve the introduction section by explaining the novelty of this review in introduction section.

4.      Please justify the novelty of this concept. In which manner this review article is different from other.

5.      Line 50: sentence is incomplete “RWS belongs to the category of effect study”

6.      Line 64: I don’t find this language scientific “research programs are often divorced from clinical practice”

7.      Line 415-416: reframe the sentence

8.      Line 510: Conclusion is not at all scientific. Kindly reframe the sentence.

9.      The authors are suggested to dually check the citations throughout the manuscript.

10.  Authors are suggested to improve the conclusion and future prospective part as per the standard format. In my opinion, authors must add future projections related to the significance of real world study in context to finding better management of human health issues. The conclusion is not supported by the content represented in the manuscript.

Author Response

Point 1: In simple summary section, Line 15-16 is not clear

Response 1: Thanks a lot for the reviewer’s carefulness and responsibility for this paper. We are not native English speakers, so please release us from the shame of any language related mistakes. According to the instructions, we have rewritten lines 15-16, which have been marked in red. Please check the revised paper.

Point 2: Grammatical and formatting issues are there in manuscript at several places

Response 2: Thanks a lot for the reviewer’s carefulness and responsibility for this paper again and we must also deeply apologize for our carelessness and fault in the primary manuscript. According to the instructions, we re-examined the grammar and format of the full text, and the corrections (lines 18-20) have been marked in red in the text. Please check the revised paper. 

Point 3: Authors are suggested to improve the introduction section by explaining the novelty of this review in introduction section.

Response 3: Thanks a lot for offering such an insightful comment. According to the instructions, we have made appropriate supplements to the introduction in lines 58-60 of the revised manuscript, and further elaborated the novelty of this review. The revision has been marked in red in the text. Please check the revised paper.

Point 4: Please justify the novelty of this concept. In which manner this review article is different from other.

Response 4: Thanks a lot for your raising such a good question. As advised, we supplement the novelty of the concept of RWS in lines 87-90 of the revised manuscript. And the differences between this review and other related reviews are further expounded in lines 61-66 of the revised manuscript. The revision has been marked in red in the text. Please check the revised paper.

Point 5: Line 50: sentence is incomplete “RWS belongs to the category of effect study”

Response 5: Thanks a lot for the reviewer’s carefulness and responsibility for this paper again. According to the instructions, we have supplemented the original sentence appropriately in lines 49-52 of the revised manuscript. The revision has been marked in red in the text. Please check the revised paper. 

Point 6: Line 64: I don’t find this language scientific “research programs are often divorced from clinical practice”

Response 6: Thank you so much for your comments. As advised, we rewrote this sentence in lines 71-72 of the revised manuscript to illustrate the shortcomings of RCT. The revision has been marked in red in the text. Please check the revised paper.

Point 7: Line 415-416: reframe the sentence

Response 7: Thank you so much for your comments again. As advised, we readjusted the language order and logic in lines 426-429 of the revised manuscript. The revision has been marked in red in the text. Please check the revised paper. 

Point 8: Line 510: Conclusion is not at all scientific. Kindly reframe the sentence.

Response 8Thanks a lot for the reviewer’s carefulness and responsibility for this paper again. As advised, after reviewing the full text, we rewrote the conclusion to make it more logical and scientific. The conclusion part has been marked in red in the text. Please check the revised paper.

Point 9: The authors are suggested to dually check the citations throughout the manuscript.

Response 9Thanks a lot for the reviewer’s carefulness and responsibility for this paper again. As advised, First, we make changes one by one in the text according to the standard reference format (reference numbers should be placed in square brackets [] and placed before the punctuation). Secondly, we checked the citations one by one throughout the manuscript. Finally, we also modify the format of the final reference of the manuscript according to the standard format.

Point 10: Authors are suggested to improve the conclusion and future prospective part as per the standard format. In my opinion, authors must add future projections related to the significance of real world study in context to finding better management of human health issues. The conclusion is not supported by the content represented in the manuscript.

Response 10Thanks a lot for your raising such a good question again. As advised,After referring to the reviews in your journal, we rewrote the conclusion and future prospective part in standard format. In this part, we first reviewed the development status and advantages of RWS. The significance of RWS and the achievements of existing RWS summarized above are further expounded. Finally, the future projection is put forward around the significance of RWS itself. This part can be supported by the original manuscript content. The revision has been marked in red in the text. Please check the revised paper. 

Besides,thanks a lot for the reviewer’s carefulness and responsibility for this paper again. Since we are not native English speakers, we have found a special "language retouching company" to retouch our review before submitting. After receiving the comments from the reviewers, we once again apologize for the carelessness and mistakes in the first draft. While correcting the language errors in the text one by one, we also asked an English-speaking colleague to review the revised manuscript.

Round 2

Reviewer 4 Report

The authors have incorporated all the suggestions in the revised manuscript. Hence, the manuscript can be accepted in its present form.